# The Hidden Charm of Life

**DOI:** 10.3390/life9010005

**Published:** 2019-01-07

**Authors:** Manuel Porcar

**Affiliations:** 1Institute for Integrative Systems Biology I2SysBio, Universitat de València-CSIC, 46980 Paterna, Spain; manuel.porcar@uv.es; 2Darwin Bioprospecting Excellence, SL., Parc Científic Universitat de València, 46980 Paterna, Spain

**Keywords:** biotechnology, synthetic biology, engineering, living system

## Abstract

Synthetic biology is an engineering view on biotechnology, which has revolutionized genetic engineering. The field has seen a constant development of metaphors that tend to highlight the similarities of cells with machines. I argue here that living organisms, particularly bacterial cells, are not machine-like, engineerable entities, but, instead, factory-like complex systems shaped by evolution. A change of the comparative paradigm in synthetic biology from machines to factories, from hardware to software, and from informatics to economy is discussed.

## 1. Synthetic Biology: On Organisms, Machines and Factories

Biotechnology is the exploitation of biological processes for industrial and other purposes. Although, in a broader sense, it encompasses classical agronomy and farming, its modern use refers to the genetic manipulation of organisms, very often microorganisms such as yeast or bacteria, for the production of useful bio-products, such as antibiotics or hormones [1]. Synthetic biology (SB) is a related field to biotechnology and metabolic engineering, which consists of an engineering view on biotechnology. There is a well-known joke among SB practitioners that stresses the lack of consensus on the definition of this emerging discipline: If one asks for a definition of SB to five synthetic biologists, one will end up with six definitions. That said, the most accepted definition is that SB consists of designing and constructing biological modules, biological systems, and biological machines in a predictive way; as well as the redesign of existing biological systems for useful purposes. As it is often summarized, SB is about making biology easier to engineer, since classical biotechnology is assumed to be mainly trial-and-error; whereas SB is expected to be—at least in the future—as reliable as any other engineering discipline.

Although the engineering ideal in biology dates back to over one hundred years ago with Jacques Loeb’s proposal on understanding life by controlling it [2], SB is indeed a young discipline: It is only less than two decades old. During this short time, though, SB has been associated with a range of milestones, such as the microbial synthesis or artemisinic acid [3], the chemical synthesis of yeast, and bacterial chromosomes (see, for example, Reference [4] or enhanced biofuel production [5]). However, many criticisms have highlighted that assay-and-error as well as tinkering or fine tuning are still massively used in SB, a discipline that, as it is the case in any other engineering branch, would be expected to be fully rationally based, straight-forward, and predictable [6]. Regardless of the relative proportion of tinkering vs. rational design, SB has provoked a real tsunami in the way bioengineering is perceived. Rather than merely using living organisms as a source of valuable products, SB aims at the redesign of natural organisms in order to convert them to à la carte sources of products and behaviors, in a metaphorically computer-, robot-, machine-like fashion. As a result of the constant analogy between living organisms and man-made constructs, SB has seen biology-to-industry metaphors flourish. It has to be stressed, though, that there has been a coexistence of the cells-as-machines metaphor—although very much criticized [7]—with a significantly different, higher heuristic value in the context of biotechnology and the cells-as-factories metaphor [8]. Rather than the obvious analogy between bacterial cells, for example, and nanorobots [9], comparative paradigms today tend to look at bacteria as microscopic, complete units of bioproduction, that is, biofactories. The change is not only a matter of scale but of complexity: A factory is much more than the sum of its robots, and its management requires a precise and yet flexible feedback with the environment.

It is interesting to highlight that the term “synthetic” in SB has two different meanings. The most common one is “artificial”, but synthesis (from the Greek σύνθεσις, composition) as opposed to analysis, also refers to the generation of complexity by the sequential addition of adjacent components. Those components (parts) can be as simple as short DNA fragments (i.e., a promoter upstream a coding sequence, ribosomal binding sites (RBS), coding sequences or terminators); or as complex as a whole cell (itself a component of a more complex system in case it is part of an artificial consortium composed of several artificial cells, for example, Reference [10]). Complexity in SB has thus been proposed to be obtained from simply combining pieces in a Lego-like fashion [11]. The hierarchy of complexity levels in SB terminology has mainly been developed in the frame of the international Genetically Engineered Machine (iGEM) competition. DNA parts are named “Biobricks” [12]. Those are assembled in “devices”, which in turn are part of the more complex “circuits”. However, this hierarchy of complexity, which is accurate enough to define a cell by analogy with an electronic apparatus or a robot, may fall short if we consider the complexity of synthetic cells, how they are expected to behave, and what are they expected to produce. Indeed, most robots, computers and other machines are “actors” in the sense that they take actions and/or produce intermediates; however, only factories—another level encompassing many of the former with a precisely defined interphase among those actors—make final products, exhibit an adaptive behavior, and interact with the environment. 

Interestingly, the classical metaphor of considering cells as machines and/or computers could be considered a reductionist view on biological complexity, since it focused on hardware and on predictability, which are at odds with the biological realm, characterized by emergent properties, a high level of noise in the biological circuits, mutation, and ultimately, evolution. Intuitively, cells seem genomically and metabolically too entangled, flexible, interconnected, and adaptable to be considered mere Lego-like complicated (not to be confounded with complex) biostructures. Moreover, the compared analysis of the Biobrick repository, Lego, and software (Java) suggests that the closest comparative framework to biological complexity should be software rather than hardware [13]. It is, thus, somehow contradictory that the exaggerated goals and hype with which SB is often described are also associated with a reductionist view on biological complexity, which is thus seen as the unwanted trait to remove. Paradoxically, the view that can better sustain the high expectations of SB is a nonreductionist approach, particularly in the case of synthetic microbiology. A nonreductionist view can support a more ambitious goal: Making cells complete, autonomous, and self-replicative biofactories that are expected to yield an almost infinite array of bioproducts. 

To finish this introduction with a last metaphor, it is tempting to me not to discuss the “first massively produced car” metaphor. Actually, and in addition its classical application to biosynthetic pathways (see, for instance, Reference [14]), the assembly line metaphor has been borrowed by structural biologists and enzymologists working on iterative biosynthetic megasynthases [15], like fatty acid synthases [16], polyketide synthases [17], and nonribosomal peptide synthases [18]. It is indeed very common in SB communications or scientific posters to display a black and white photograph of a Ford “T” model (produced between 1908 and 1927) being assembled in a beautifully simple and organized assembly line. This image is almost always used as a proxy of what synthetic biology should be. However, at least in this author’s opinion, the precise nature of analogy is different, since it should be stressed that the comparative framework for a synthetic cell is not a car being produced in an assembly line of a factory, but the factory itself. 

## 2. What Is Life? On Pillars and Sizes

It has been suggested that life has seven pillars, which are program, improvisation, compartmentalization, energy, regeneration, adaptability, and seclusion (isolation) [19]. That said, these features can be summarized in three: Metabolism, self-reproduction, and evolution. It is indeed difficult to envisage a living organism without those three traits. Viruses are a special case, since although they clearly—and rapidly—evolve, their parasitic nature makes them only replicable in a cellular level—the victim host–environment, whose metabolism is partly deviated to create the new viral progeny [20]. However, if we keep the focus, as it is almost always the case in SB, on bacterial cells, it is evident how the metabolism drives adaptation and, thus, reproduction and evolution [21]. In his provocative abovementioned assay, de Lorenzo suggested that “the interplay of DNA and metabolism is in my view akin to that of politics and economy. Both realms drive their own autonomous agendas and obviously influence each other. But whether one likes it or not, it is economy that ultimately determines the viability of any political move. By the same token, metabolism (i.e., the economy of living systems) frames and ultimately resolves whether a given genetic program (i.e., already existing, knocked-in by horizontal gene transfer or engineered with recombinant DNA technology) can be deployed or not” [21]. Interestingly, the abovementioned three pillars are irremediably interconnected: Metabolism is the central functional core of a cell (with a bidirectional feedback with the genome), and the interphase of the metabolism with the environment and the laws of physics, in time, will determine the evolutionary success of the genome and its variants through evolution. 

Physics- and chemistry-centered implications have been used to try to shed light on the very essence of life. On the former, Schrödinger’s view is by far the most well-known. In 1944, he wrote his famous book *What is life? The Physical Aspect of the Living Cell* [22]. In his book, Schrödinger highlights the importance for organisms to be big enough (significantly bigger than atoms) in order for them to escape from the randomness of the circa-atomic world (i.e., Brownian motion). However, being large compared to atoms can still mean being very small compared to machines. Indeed, the size of cellular and multicellular living beings varies from a tiny *Mycoplasma* spp. (10^−13^ g) to the largest animal, the blue whale (10^9^ g). This means that the variation in size from the smallest to the largest cell-based organism is as much as 10^22^. 

The machine vs. cell metaphor is always discussed in terms of function. However, it is interesting to address the analogy by also considering the size of both structures. Machines are typically in the scale of g to kg, which is interestingly in the same range of the most abundant (by mass) organisms on Earth: Plants. However, around one fourth of the biomass on Earth [23], and the vast majority of it in terms of biodiversity, is microscopic, which is in contrast with the poorly developed human-made equivalents, such as nanomachines and nanorobots. Some machines can be huge, compared to organisms. The largest machine, the Large Hadron Collider, LHC (https://www.nature.com/news/2008/080905/full/news.2008.1085.html) measures 27 km and weighs around 88,000 tones (that is, around 500-fold larger than a blue whale). On the other hand, the smallest nanomachines have been designed at the molecule scale [24], smaller that viruses. However, these devices exhibit a very simple behavior, are still in their infancy, and often depend on external inputs from a extremely bigger machine. As a conclusion of this discussion of sizes, the one of both machines (and factories) and living organisms is globally relatively similar, although it encompasses a wide range of variation. However, machines exhibit a clear preference for the macroscopic world whereas living organisms have a clear taste for the irregular and unordered (according to Schrödinger) microscopic scale. Machines are relatively large structures because it is easier to assemble macroscopic structures, but also because a nonmicroscopic device is more amenable to a mechanical/electronic design. Tiny objects—such as cells—are ruled by complex chemical (metabolic) reactions, in addition to mechanical forces, which, although it does not mean that they are not engineerable, certainly poses an additional challenge for them to be the subject of classical industrial design.

Schrödinger’s book has received many criticisms for wrongly attributing the genetic basis of inheritance to proteins, for his proposal of “negative entropy” to explain metabolism, and, particularly, for his mystic proposals in the Epilogue [25]. Yet, the book has had an important influence on biologists and physicists. Maybe part of the charm of the book was its arguable conclusion that new laws of physics are required to explain living matter. In his famous assay “Life’s Irreducible Structure”, published in *Science* in 1968, Polanyi extensively used the analogy between organisms and machines, having both, in his words, “mechanisms”, which were “boundary conditions harnessing the laws of inanimate nature, being themselves irreducible to those laws” [26]. This brings us back to the classical question: Are living beings just a particular kind of machine that makes more machines? [27] This issue is central in SB, since the whole discipline is based on the biomachine analogy and takes it for granted that, therefore, living beings *must* be engineerable even if, as we have seen above, they tend to be significantly smaller than actual machines. It is, thus, a paradox that synthetic biology has chosen tiny cells as the best proxy for machines, considering that autonomous, actual machines are never that small. In other words, cells are out of the range of sizes for a complex machine, let alone a factory. Current progress in both SB and nanorobotics may help to close this gap in the size between cells and machines. 

## 3. Why Life? On the Choice of Organisms for Engineering

The question “why life?” hopelessly evokes the hard issue of the origin of life on our planet, which is well beyond the scope of this manuscript. However, the question is also relevant when it comes to analyzing why biology has been chosen as a subject. The field of SB, with a clear engineering origin, seems to have an ambiguous approach on the main traits of the living realm. On one hand, evolution and emergent properties are considered the enemy to defeat in order for rational design to be predictable. On the other hand, evolution and natural selection are seen as useful tools that can contribute to getting or to optimizing designs [28]. However, in addition its use as a tool, is evolution a desired or undesired trait? In order to answer this question, we first need to consider another one: Is life possible without evolution? There is a clear link between the intensity of selection with the stability of the environment. The more stable the environment, the less (further) adaptation it requires. However, even the most stable environment one can consider (a laboratory fermenter, with a precisely defined medium, inoculated with a pure culture and operated under strictly controlled conditions, for example) is a scenario in which mutation, selection, genetic drive, and thus, evolution, occur. An example of this is the unwanted selection of biofilm-makers and/or cheaters during industrial fermentations. Evolution is the essence of life but a pain in the neck for rational designers. Is it really needed? Some think that we can get rid of evolution. A few years ago, I was mentoring a team of students attending the international Genetic Engineered Machine (iGEM) competition. The project, Talking Life (http://2012.igem.org/Team:Valencia_Biocampus), consisted of a system that allowed to communicate with engineered bacteria through a simple yet ingenious strategy. The approach started by converting oral questions into light pulses that would specifically excite sensitive fluorescent proteins, which would only be present under certain conditions. For example, by placing a gene encoding a fluorescent protein under the control of a heat-sensitive promoter, one could know the thermal comfort of the cell population by translating the question “are you hot?” into a wavelength that would excite the cognate protein, which would then emit light in another wavelength, which would be translated into the “bacterial answer”: “Yes, I am”. However, things did not work as expected. In the hot summer in Valencia, Spain, where the experiments took place, an electric failure disconnected the refrigeration system of an orbital shaker containing the engineered *E. coli* strains, which were thus subjected to a high heat stress for several hours. The day after, the culture was exposed to a controlled, short heat shock and then “asked” whether it was hot, by means of the strategy described above. The answer to the question was “no”, even if it should be “yes”: Unexpectedly, there was no expression of the fluorescent protein (although the media were indeed hot). The reason was that the unintended heat stress that the culture had suffered had resulted in the selection of a population of cheaters, which were yielding the “wrong” answer [29]. Paradoxically enough, though, the first attempt to communicate with bacteria ended up with synthetic microorganisms lying.

## 4. Conclusions: The Hidden Charm of Life

*The Purloined Letter* by Edgar Allan Poe [30] features a beautifully written short story in which a very important object—a letter—is unsuccessfully searched by the Paris Police. Finally, detective Dupin finds it: It had always been so visible that no one could thing that that object was the letter the police were looking for.

SB is and engineering discipline working on a matter recalcitrant to be engineered. Maybe this explains the success of all kinds of metaphors on living cells [31], which are almost always compared with machines [32]—although biofactories could be a better choice, as we have discussed; on SB itself, which is compared with hardware-building or Lego (software may be a more precise analogy); and, finally, metabolism, which is correctly compared with the economy [21]. Artificial devices follow the laws of physics, and so do biological organisms, but the particularities of the organization of the living (metabolism, reproduction, evolution) are not shared with machines, generally speaking. There are very interesting examples, though, of machines producing other machines or, even, machines amenable to evolution [33,34]. This leads us to the surprising possibility that life-like creation may be achieved through robotics and AI before cellular-based SB fully succeeds. If an artificial machine had software to interact with the environment, if it was self-replicative, and the machine’s offspring was more or less viable depending on evolutionary forces…would that machine be alive?

Living beings are more than machines that build machines (in fact, they are closer to factories that make factories) and it has to be stressed that the central pillar of life is not a material one but a process. SB is not about designing organisms, but about taming such a process: Evolution, the Purloined Letter of SB.

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
