# Peer review of "The Hidden Charm of Life"

_life, 2019, doi:10.3390/life9010005_

Round 1
Reviewer 1 Report
This Perspective is well designed and written - I enjoyed reading it. It is of interest to the synthetic biology community, where these debates on how to communicate and frame the field (hardware vs software, machine vs factory...) are important.
I have some minor comments:
Abstract says that SB "is expected to revolutionise" - hasn't it already?. Similar comment at the very end (line 202): "...before cellular-based SB succeeds". I think it has succeed already, to some extent at least.
Line 69: "the metaphor of considering cells as computers is in fact a very reductionist view on biological complexity". I don't agree here - the author might be referring to electronic circuits. That metaphor could be reductionist. By computer I understand a device in which both hardware and software interact in many complex ways. I do not see the reductionist view in this metaphor (it could well be the case, so please elaborate a bit).
Line 82: "the author of this article". Line 169: "I". 3rd or 1st?
I do not understand the size comparison. I just don't see the point, so please clarify. Line 148: "It is thus a paradox that synthetic biology has chosen tiny cells as the best proxy for machines, considering that actual machines are never that small". I always thought that the machine vs cell metaphor was in terms of function (a behavioural comparison, or a computing comparison if you wish) rather than a size comparison.
The author establishes a very good point with the hardware vs. software issue. However, this discussion disappears when dealing with evolution (line 153 onward). Genetic programming (with genetic algorithms) and evolutionary programming (among others) are software methods in which evolution is somewhat mimicked for the benefit of the problem to solve. I guess this discussion is out of the scope of this manuscript, but I wonder if the use of evolution (kind of) by software engineers/developers could help synthetic biologists here.
Author Response
This Perspective is well designed and written - I enjoyed reading it. It is of interest to the synthetic biology community, where these debates on how to communicate and frame the field (hardware vs software, machine vs factory...) are important.
Thank you for your comments.
I have some minor comments:
Abstract says that SB "is expected to revolutionise" - hasn't it already?. Similar comment at the very end (line 202): "...before cellular-based SB succeeds". I think it has succeeded already, to some extent at least.
Both sentences have been slightly modified in the revised version. The former now reads “has alreadyrevolutionized” and the latter now reads “(…) before cellular-based SB fullysucceeds”.
Line 69: "the metaphor of considering cells as computers is in fact a very reductionist view on biological complexity". I don't agree here - the author might be referring to electronic circuits. That metaphor could be reductionist. By computer I understand a device in which both hardware and software interact in many complex ways. I do not see the reductionist view in this metaphor (it could well be the case, so please elaborate a bit).
I understand R1 point, but I personally do believe that considering cells as computers is a reductionist view (the adjective “very” has been removed in the revised version). However, I agree with the reviewer that the statement was too sharp and needed further clarification. The revised sentence now reads (changes in bold) “Interestingly, the classical metaphor of considering cells as machines and/or computers could beconsidered a reductionist view on biological complexity, since it makes a focus on hardware and on predictability, which are at odds with the biological realm, characterized by emergent properties, high level of noise in the biological circuits, mutation, and, ultimately, evolution.” References 6 and 7, cited elsewhere in the manuscript, support this (personal) view which, as R1 says, could well be wrong.
Line 82: "the author of this article". Line 169: "I". 3rd or 1st?
I am using 1st person in both cases (Line 169 now reads “me”)
I do not understand the size comparison. I just don't see the point, so please clarify.
Line 148: "It is thus a paradox that synthetic biology has chosen tiny cells as the best proxy for machines, considering that actual machines are never that small". I always thought that the machine vs cell metaphor was in terms of function (a behavioural comparison, or a computing comparison if you wish) rather than a size comparison.
I fully agree: the machine vs cell metaphor has always been used in terms of function. Because of that, in this manuscript I wanted to explore the “size factor”, which has not been discussed to my knowledge in the SB debate. My point is as follows. If we want to build bio-machines (so to speak), how small can they be? Cells are tiny, machines are not, and the fact that most living beings are microscopic is never considered as a problem. Interestingly, though, machines are relatively large (typically in the cm-m range) because it is easier to assemble macroscopic structures but also because a non-microscopic device is more amenable to a mechanical/electronic design. Tiny objects –such as cells- are more “chemical than physical”, which although does not mean that they are not engineerable, is an additional challenge for their “machinisation”.
The reviewer is right on the lack of clarity of this argument in the original manuscript, which has been modified to include (more formally) the discussion I have made above. (Line 127-147).
The author establishes a very good point with the hardware vs. software issue. However, this discussion disappears when dealing with evolution (line 153 onward). Genetic programming (with genetic algorithms) and evolutionary programming (among others) are software methods in which evolution is somewhat mimicked for the benefit of the problem to solve. I guess this discussion is out of the scope of this manuscript, but I wonder if the use of evolution (kind of) by software engineers/developers could help synthetic biologists here.
Yes, that’s right. The discussion is out of the scope of this piece, but please notice that a mention on the evolution-based strategies that are indeed used in the SB framework is included in the manuscript, as follows: “On the other side, evolution and natural selection are seen as useful tools that can contribute to get or to optimize designs [28].”
Reviewer 2 Report
In this perspective, Manuel Porcar gives food for thought by reflects on life through the lens of synthetic biology. The piece is very well written, the content is almost lyrical, and my opinion is that we need articles like this in the field. As the author mentions, "if one asks for a definition of SB to five synthetic biologists, one will end up with six definitions". I thus may not fully agree with all the points and the metaphors, but as a reviewer I find the arguments well-supported. I therefore recommend the article to be accepted for publication.
I have three points that the author might want to include or consider for the final version:
There are a few works on the use of metaphors and their use in synthetic biology (e.g. https://lsspjournal.biomedcentral.com/articles/10.1186/s40504-017-0061-y https://lsspjournal.biomedcentral.com/articles/10.1186/s40504-018-0077-y) Maybe the author would like to trefer the reader to them to complement the discussion?
The author deos not refer to the research done to produce a minimal bacterial cell and the synthetic chromosomes in yeast. Aren't those two works central to answering big biological questions in synthetic biology?
In relation to the previous point, synthetic biology has tremendous use in fundamental research (understanding by synthesis). Although there is a quick mention, I believe that, given the scope of the article, this point deserves some more attention
Author Response
In this perspective, Manuel Porcar gives food for thought by reflects on life through the lens of synthetic biology. The piece is very well written, the content is almost lyrical, and my opinion is that we need articles like this in the field. As the author mentions, "if one asks for a definition of SB to five synthetic biologists, one will end up with six definitions". I thus may not fully agree with all the points and the metaphors, but as a reviewer I find the arguments well-supported. I therefore recommend the article to be accepted for publication.
Thank you very much for your kind comments.
I have three points that the author might want to include or consider for the final version:
There are a few works on the use of metaphors and their use in synthetic biology (e.g. https://lsspjournal.biomedcentral.com/articles/10.1186/s40504-017-0061-y https://lsspjournal.biomedcentral.com/articles/10.1186/s40504-018-0077-y) Maybe the author would like to refer the reader to them to complement the discussion?
Thank you. Both references are now included in the revised manuscript.
The author does not refer to the research done to produce a minimal bacterial cell and the synthetic chromosomes in yeast. Aren't those two works central to answering big biological questions in synthetic biology?
Yes, they are. Their absence is premeditated. There are many review and perspective articles either hyping or strongly criticizing those milestones. I have done it in the past as well. But from the integrative knowledge point of view, their contribution to the field is not that important: they have served to i) let us know what can be build and ii) how poor is still our knowledge on the biological interactions and genome/metabolism complexity.
In relation to the previous point, synthetic biology has tremendous use in fundamental research (understanding by synthesis). Although there is a quick mention, I believe that, given the scope of the article, this point deserves some more attention.
I see the reviewer point, and certainly the famous quote by R. Feynmann “what I cannot build I cannot understand” goes in this direction. But as stressed above, I did not want the manuscript to focus on the knowledge-towards-synthesis debate but rather concentrate on the features of live as they relates to “machinisation”.
Reviewer 3 Report
The authors should include recent advancements in the field of understanding the basics of life, utilizing the knowledge for betterment of life and tuning of life functions for designing designers' life using techniques as artificial chromosome, multiplexed regulations and gene editing. Besides they can include recent breakthroughs by AIs and big data analysis for understanding the life at molecular level or designing artificial life components as artificial organs etc.
Some references that can be included:
https://www.nature.com/articles/d41586-018-07607-3
https://www.proteinatlas.org/
https://www.humancellatlas.org/
https://www.nature.com/articles/s41586-018-0518-z
https://www.nature.com/articles/nbt.4137
https://dx.doi.org/10.1021/acssynbio.7b00259
https://aabme.asme.org/categories/artificial-organs
And many more!
Author Response
The authors should include recent advancements in the field of understanding the basics of life, utilizing the knowledge for betterment of life and tuning of life functions for designing designers' life using techniques as artificial chromosome, multiplexed regulations and gene editing. Besides they can include recent breakthroughs by AIs and big data analysis for understanding the life at molecular level or designing artificial life components as artificial organs etc.
Some references that can be included:
1.-https://www.nature.com/articles/d41586-018-07607-3
2.-https://www.proteinatlas.org/
3.-https://www.humancellatlas.org/
4.-https://www.nature.com/articles/s41586-018-0518-z
5.-https://www.nature.com/articles/nbt.4137
6.-https://dx.doi.org/10.1021/acssynbio.7b00259
7.-https://aabme.asme.org/categories/artificial-organs
Thank you for your comments. I understand your point and share your view on the immense contribution to the field of SB/Biotechnology/biomedicine of all the references you mention, and which I have read or re-read upon your advice.
However, please notice that a methodology-focus view is not the approach I am following in this manuscript. My article is not a review on SB-related techniques, but a personal (and thus opinionated) article on some of the features of living beings that are to be considered as a conceptual starting point for SB. The above references you suggest to cite, refer to CRISPR techniques in yeast or mice (refs 1,5,6, above), to Human proteins/cells (2, 3, 4) or to artificial organs (7). While being of enormous interest for a review on the last techniques SB can use for genetic engineering in eukaryotes, all those references are at the other extreme of the SB landscape I am exploring in this paper.
Round 2
Reviewer 3 Report
I have consent for publication in current form after revision from the authors.